# Structured Uncertainty in the Observation Space of Variational Autoencoders

## Abstract

Variational autoencoders (VAEs) are a popular class of deep generative models with many variants and a wide range of applications. Improvements upon the standard VAE mostly focus on the modelling of the posterior distribution over the latent space and the properties of the neural network decoder. In contrast, improving the model for the observational distribution is rarely considered and typically defaults to a pixel-wise independent categorical or normal distribution. In image synthesis, sampling from such distributions produces spatially-incoherent results with uncorrelated pixel noise, resulting in only the sample mean being somewhat useful as an output prediction. In this paper, we aim to stay true to VAE theory by improving the samples from the observational distribution. We propose an alternative model for the observation space, encoding spatial dependencies via a low-rank parameterisation. We demonstrate that this new observational distribution has the ability to capture relevant covariance between pixels, resulting in spatially-coherent samples. In contrast to pixel-wise independent distributions, our samples seem to contain semantically meaningful variations from the mean allowing the prediction of multiple plausible outputs with a single forward pass.

## 1 Introduction

Generative modelling is one of the cornerstones of modern machine learning. One of the most used and widespread classes of generative models is the Variational Autoencoder (VAE) (Kingma & Welling, 2014; 2019). VAEs explicitly model the distribution of observations by assuming a latent variable model with low-dimensional latent space and using a simple parametric distribution in observation space. Using a neural network, VAEs decode the latent space into arbitrarily complex observational distributions.

Despite many improvements on the VAE model, one often-overlooked aspect is the choice of observational distribution. As an explicit likelihood model, the VAE assumes a distribution in observation space – using a delta distribution would not allow gradient based optimisation. Most current implementations, however, employ only simple models, such as pixel-wise independent normal distributions, which eases optimisation but limits expressivity. Else, the likelihood term is often replaced by a reconstruction loss – which, in the case of an $L_2$ loss, implicitly assumes an independent normal distribution. Following this implicit assumption, samples are then generated by only predicting the mean, rather than sampling in observation space.

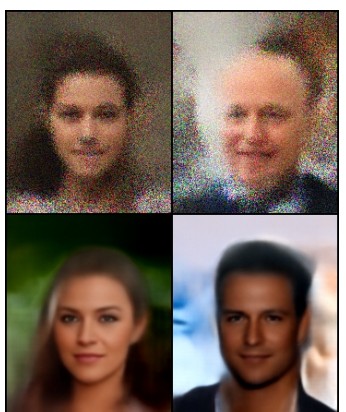

Figure 1: **Top:** Samples generated with a standard VAE exhibiting pixel-wise independent noise. **Bottom:** Samples with our structured observation space VAE are realistic and spatially coherent.

An application where this disconnect becomes apparent is image synthesis. The common choices for observational distributions are pixel-wise independent categorical or normal distributions. For pixel-wise independent distributions, regardless of other model choices, sampling from the joint distribution over pixels will

result in spatially-incoherent samples due to independent pixel noise (cf. Figure 1). To address this problem, researchers use the predicted distributions to calculate the log-likelihood in the objective but then discard them in favour of the mean when generating samples or reconstructing inputs. However, this solution is akin to ignoring the issue rather than attempting to solve it.

In this work, we explore what happens when we strictly follow VAE theory and sample from the predicted observational distributions. We illustrate the problem of spatial incoherence that arises from using pixel-wise independent distributions. We propose using a spatially dependent joint distribution over the observation space and compare it to the previous scenario. We further compare the samples to the mean of the predicted observational distribution, which is typically used when synthesising images. We note that, in this work, we are not focusing on absolute image quality. Instead, we aim to point to an issue often overlooked in VAE theory and application, which affects most state-of-the-art methods. Thus we analyse the relative difference between using and not using a joint pixel dependent observational distribution for a basic VAE. Yet, our findings are of broad relevance and our proposed model can be used in more advanced VAE variants.

## 2 RELATED WORK

Modern generative models can be divided into two classes, implicit likelihood models, such as Generative Adversarial Networks (GANs) (Goodfellow et al., 2014) and diffusion models (Song & Durkan, 2021), and explicit likelihood models, such as VAEs (Kingma & Welling, 2014; Rezende et al., 2014; Kingma & Welling, 2019), flow models (Dinh et al., 2017; Kingma & Dhariwal, 2018; Dinh et al., 2015) and auto-regressive models (Van Den Oord et al., 2016a;b; Salimans et al., 2017). Despite implicit likelihood models having achieved impressive results in terms of sample quality without the need for explicitly modelling the observation space (Karras et al., 2020; Brock et al., 2019), interest in explicit likelihood models have prevailed due to their appealing properties, such as ease of likelihood estimation.

One of the most popular and successful explicit likelihood models is the VAE (Kingma & Welling, 2014; Rezende et al., 2014; Kingma & Welling, 2019). Since its introduction, there have been numerous extensions. For example, Van Den Oord et al. (2017); Razavi et al. (2019) quantize the latent space to achieve better image quality; Higgins et al. (2017); Chen et al. (2017) modify the latent posterior to obtain disentangled and interpretable representations; and Vahdat & Kautz (2020); Sønderby et al. (2016) use hierarchical architectures to improve sample quality.

Like most other explicit likelihood models, the VAE requires the choice of a parametric observational distribution. This choice is often pixel-wise independent. As a result, practitioners use the distribution to calculate the likelihood but use its expected value when sampling, as the samples themselves, are noisy and of limited use in most applications. However, according to the theory and as previously pointed out by Stirn & Knowles (2020) and Detlefsen et al. (2019), a latent space sample should entail a distribution over observations and not a single point. Despite attempts to enforce spatial dependencies in the decoder architecture (Miladinovic et al., 2021), without a pixel-dependent joint likelihood, the observation samples will remain noisy. Notable exceptions are auto-regressive models and auto-regressive VAE decoders (Van Den Oord et al., 2017; Razavi et al., 2019; Gulrajani et al., 2017; Nash et al., 2021). Unlike other explicit likelihood models, auto-regressive models jointly model the observational distribution by sequentially decoding pixels while conditioning on previously decoded values. While this sampling procedure results in spatially coherent samples, it is computationally expensive and uncertainty estimation is not trivial.

In the context of non-auto-regressive VAE decoders, work focusing on modelling a joint observational distribution that accounts for pixel dependencies is limited. Monteiro et al. (2020) use a low-rank multivariate normal distribution to produce spatially consistent samples in a segmentation setting. However, they focus on discriminative models only. For generative models, a notable exception is a work by Dorta et al. (2018) which, similarly to our proposed method, employs a non-diagonal multivariate normal distribution over observation space. The key difference is the choice of parameterisation used for the covariance matrix. Dorta et al. (2018) predict the Cholesky-decomposed precision matrix, which grows quadratically with the size of the image. To address this computational constraint, the authors use a sparse decomposition. This decomposition considers only a local neighbourhood of pixels, which limits its ability to capture long-range spatial dependencies. In contrast, our approach uses a global parameterisation.

## 3  METHODS

### 3.1  VARIATIONAL AUTOENCODERS

We briefly revisit the theory of standard VAEs as proposed by Kingma & Welling (2014); Rezende et al. (2014). We assume a prior distribution over the latent variables, $p(\mathbf{z})$, a probabilistic encoder of the posterior, $p_{\boldsymbol{\theta}}(\mathbf{z}|\mathbf{x})$, and a probabilistic decoder of the likelihood, $p_{\boldsymbol{\theta}}(\mathbf{x}|\mathbf{z})$. In practice, the probabilistic encoder describes an intractable posterior. Using variational inference, we approximate this posterior with a distribution, $q_{\boldsymbol{\phi}}(\mathbf{z}|\mathbf{x})$ given parameters $\boldsymbol{\phi}$. Here, VAEs provide the algorithm to jointly learn the parameters $\boldsymbol{\theta}$ and $\boldsymbol{\phi}$ (Kingma & Welling, 2014). Considering some dataset $\mathbf{X} = \{\mathbf{x}^{(i)}\}_{i=1}^{N}$, the VAE objective is given by maximising the evidence lower bound with respect to the parameters $\boldsymbol{\theta}$ and $\boldsymbol{\phi}$ (Bank et al., 2020; Kingma & Welling, 2014):

$$\mathcal{L}(\boldsymbol{\theta}, \boldsymbol{\phi}; \mathbf{x}^{(i)}) = -D_{KL}\left[q_{\boldsymbol{\phi}}(\mathbf{z}|\mathbf{x}^{(i)})||p_{\boldsymbol{\theta}}(\mathbf{z})\right] + \mathbb{E}_{q_{\boldsymbol{\phi}}(\mathbf{z}|\mathbf{x}^{(i)})}\left[\log p_{\boldsymbol{\theta}}(\mathbf{x}^{(i)}|\mathbf{z})\right] \tag{1}$$

Since the derivative of the lower bound w.r.t $\boldsymbol{\phi}$ is problematic due to the stochastic expectation operator, the re-parameterisation trick is used to yield the Stochastic Gradient Variational Bayes (SGVB) estimator (Kingma & Welling, 2014).

Note, from the second term in equation 1, that the optimisation objective requires the choice of an observational distribution to calculate the likelihood that the data comes from the predicted distribution: $p_{\boldsymbol{\theta}}(\mathbf{x}|\mathbf{z})$. Hence, it follows that a latent sample entails a distribution over observations. The predicted distribution should be as close as possible to the observed distribution. Its samples should look like real observations. However, in the case of highly structured data such as images, this will not be the case when using the commonly-employed pixel-independent joint distribution.

### 3.2  STRUCTURED OBSERVATION SPACE VARIATIONAL AUTOENCODERS

With the commonly assumed model for the observation space, where the predicted joint distribution is pixel-wise independent, samples can only add noise to the predicted mean, as shown in the examples in section 4.1. By incorporating spatial dependencies in the predicted distribution, we aim to overcome this limitation and generate more realistic samples under the observed distribution.

Our solution is to replace the joint independent distribution predicted by the decoder with one that explicitly models dependencies between outputs. Specifically, we modify the final layer of the decoder to predict a low-rank parameterisation of a fully populated covariance matrix for use in a multivariate normal distribution, $p_{\boldsymbol{\theta}}(\mathbf{x}|\mathbf{z}) \sim \mathcal{N}(\boldsymbol{\mu}, \boldsymbol{\Sigma})$. This small modification can be applied to most existing VAE architectures.

Inspired by a recent discriminative model (Monteiro et al., 2020), we use an efficient parameterisation, $\boldsymbol{\Sigma} = \mathbf{P}\mathbf{P}^{T} + \mathbf{D}$, to model covariance globally, albeit at a low-rank. This yields a compact model with a covariance factor, $\mathbf{P} \in \mathbb{R}^{(S \times C) \times R}$ and a covariance diagonal, $\mathbf{D} = \mathrm{diag}(\mathbf{d}) \in \mathbb{R}^{(S \times C)^2}$, with diagonal elements, $\mathbf{d}$. Here, $S = H \times W$ is the number of pixels, $C$ is the number of channels, and $R$ is the rank of the parameterisation. Since we have only modified the distribution of the likelihood, the SGVB estimator (equation 1) is applicable without modification.

However, we found optimising the SGVB estimator with a non-diagonal covariance to be unstable regarding the variance. The instability comes from the fact that two routes can optimise the likelihood: to find the correct mean and appropriate variance around it or to keep increasing the variance (uncertainty about the mean). The second direction is obviously undesirable and results in implausible samples (with overly bright colours and high contrast; cf. Appendix A). Monteiro et al. (2020) observed a similar phenomenon and pre-trained on the mean to avoid it. This problem is not unique to our implementation; the stability of variance networks has been discussed before (Stirn & Knowles, 2020; Detlefsen et al., 2019).

In our case, we found pre-training the mean to be beneficial but insufficient. To further mitigate the problem, our solution includes weight initialisation, fixing the covariance diagonal to a small positive scalar: $\mathbf{D} = \epsilon \mathbf{I}$ [1], and constraining the entropy of the predicted distribution. Constraining the entropy constrains the variance indirectly, thus giving preference to low-variance solutions. We compute

---

[1] We found $10^{-5}$ to yield good results.

the entropy of the normal distribution in closed form and add it to the objective function. We employ soft-constraints for the entropy and KL divergence using the modified differential method of multipliers (Platt & Barr, 1988), for its agreeable convergence and stability properties, which results in the following Lagrangian formulation:

$$\tilde{\mathcal{L}}(\boldsymbol{\theta}, \boldsymbol{\phi}; \mathbf{x}^{(i)}) = \frac{1}{M} \sum_{m=1}^{M} \log p_{\boldsymbol{\theta}}(\mathbf{x}^{(i)}|\mathbf{z}^{(i,m)}) \tag{2}$$
$$- \beta \left[ D_{KL} \left[ (q_{\boldsymbol{\phi}}(\mathbf{z}|\mathbf{x}^{(i)})||p_{\boldsymbol{\theta}}(\mathbf{z})) \right] - \xi_{KL} \right]$$
$$- \lambda_H \left[ H(\mathbf{x}^{(i)}|\mathbf{z}) - \xi_H \right]$$

where $\beta$ and $\xi_{KL}$ are the Lagrangian multiplier and the slack variable, respectively, for the $\beta$-VAE constraint; $H(\mathbf{x}^{(i)}|\mathbf{z})$ is the entropy of the predicted distribution and $\lambda_H$ and $\xi_H$ are the Lagrangian multiplier and the slack variable, respectively, for the new constraint.

## 4 EXPERIMENTS AND RESULTS

### 4.1 STANDARD VAE VS. STRUCTURED OBSERVATION SPACE VAE

We start by comparing a standard VAE, with a pixel-wise independent normal observational distribution, to the proposed method, with a low-rank multivariate normal observational distribution. We perform the comparison in two datasets: the CELEBA dataset (Liu et al., 2015) and the UK Biobank (UKBB) Brain Imaging dataset (Miller et al., 2016). For all models, we use a latent space of dimension 128. and a target KL loss. For the low-rank model, we use a rank of 25. For the CELEBA dataset we use a target KL loss , $\xi_{KL}$, of 45 for both models and $\xi_H = -504750$ for our model. For the UKBB dataset we use a target KL loss , $\xi_{KL}$, of 15 for both models and $\xi_H = -198906$ for our model. Figures 2a & 2b and 2c & 2d show the qualitative results for the comparison.

In Figures 2a and 2c, we see that the samples of the standard VAE exhibit uncorrelated pixel noise around the mean, resulting from the pixel-wise independent joint observational distribution. In contrast, in Figures 2b and 2d, we see that the samples produced by our method contain semantically meaningful variations around the mean and are spatially coherent, as illustrated in the difference (row 3) between the mean (row 1) and the sample (row 2). Looking at the variance of the two methods (row 4), we see a significant difference in the regions where each model is uncertain, highlighting the difference in behaviour between the two predicted distributions. The predicted covariance (rows 5 and 6) for the low-rank model contains a structure that pertains to the image content. These figure rows represent positive and negative covariance to the central pixel indicating global covariance can is modelled. This structure results in spatially coherent samples as opposed to the noisy samples of the standard VAE, which are a consequence of the diagonal covariance. Interestingly, we observe more variation in the means of the standard VAE, suggesting that as more variation can be modelled in the observation space, less needs to be modelled in the latent space.

Quantitative evaluation of generative modelling is an inherently difficult task due to its subjective nature. While measuring the log-likelihood is the obvious choice, it is often not indicative of sample quality (Theis et al., 2016; Borji, 2019). The Fréchet Inception Distance (FID) (Heusel et al., 2017) is the current standard choice of metric due to its consistency with human perception (Borji, 2019). We note this metric is not without its criticisms (Borji, 2019; Razavi et al., 2019), regardless, we use it to evaluate our generative models and report the results in Table 1. The results do not represent the absolute performance of the proposed method, but rather the relative difference when compared to a standard VAE with a pixel-wise independent observational distribution while everything else is constant. We emphasise, the proposed method is compatible with generative models from other works. We observe that, for both datasets, the proposed method achieves a lower FID score than the standard VAE. Notably, the samples from the model with a low-rank multivariate normal observational distribution outperform the means of a standard model, indicating that sampling from the observational distribution, as theory entails, does not reduce image quality.

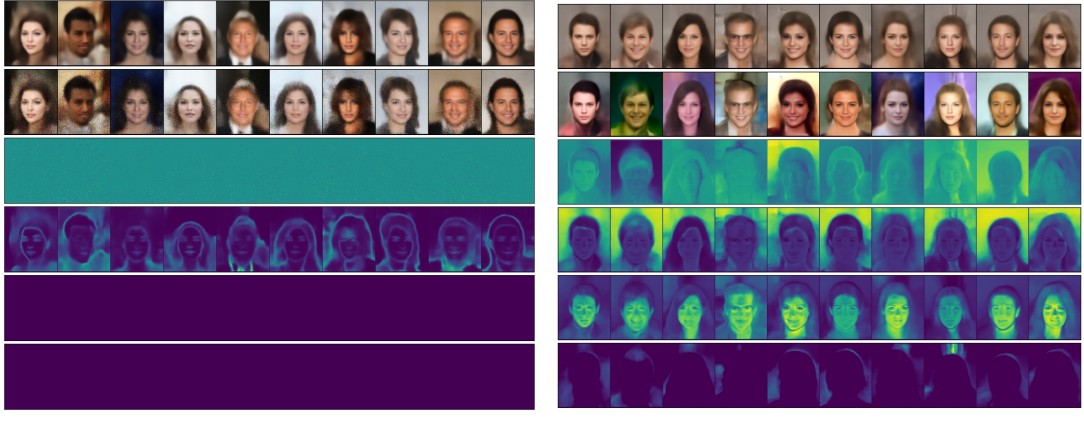

(a) Standard VAE on CELEBA           (b) Structured Observation Space VAE on CELEBA

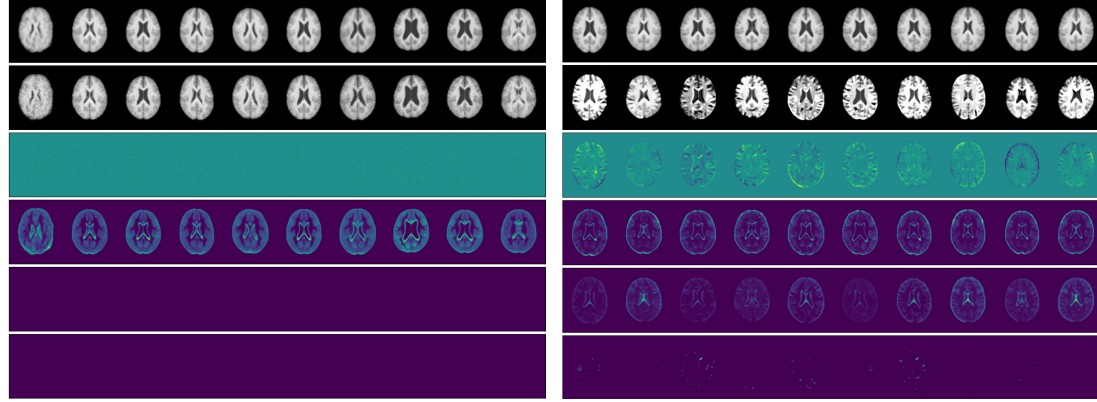

(c) Standard VAE on UKBB Brain Scans      (d) Structured Observation Space VAE on UKBB

Figure 2: Qualitative results comparing a standard VAE (2a) and the proposed VAE (2b) on the CELEBA dataset. Same comparison on the UKBB dataset (2c & 2d). The rows from top to bottom: the mean of the observational distribution, a sample from the observational distribution, the difference between the mean and the given sample, the pixel-wise independent variance per pixel, a slice of the covariance matrix: positive covariance and negative covariance to the central pixel.

Table 1: FID metric results for a standard VAE with a pixel-wise independent observational distribution and our modified VAE. Lower FID scores represent better performance.

| Method | Dataset | FID ↓ |
|---|---|---|
| Standard VAE (means) | | 121.65 |
| Standard VAE (samples) | CELEBA | 196.40 |
| Our VAE (means) | | 132.93 |
| Our VAE (samples) | | **104.62** |
| Standard VAE (means) | | 211.24 |
| Standard VAE (samples) | UKBB | 332.89 |
| Our VAE (means) | | 141.87 |
| Our VAE (samples) | | **79.712** |

## 4.2 INTERPOLATION IN THE OBSERVATION SPACE

To explore the expressiveness of the representations captured in the observation space, we visualise a continuous range of samples from the proposed method. We perform spherical linear interpolation over the observation space to capture a range of plausible images between two initial samples,

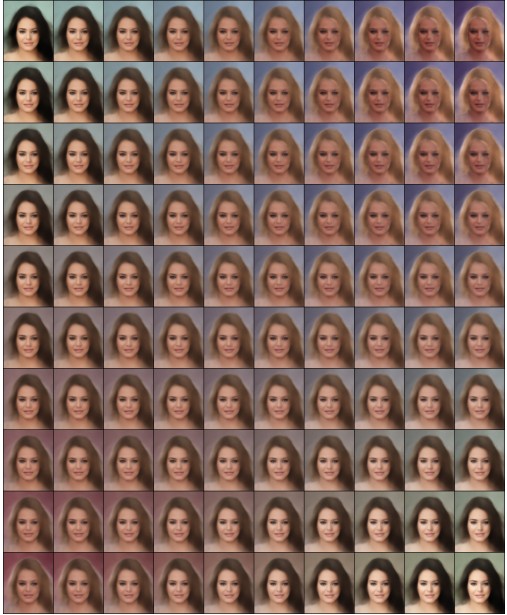 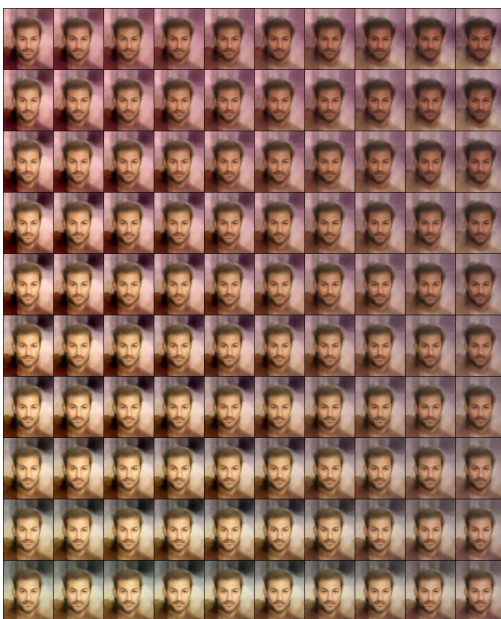

Figure 3: Spherical linear interpolation between auxiliary noise variables $\boldsymbol{\omega}_p$. The four corners are random samples from a predicted distribution with all intermediate steps as interpolations between them. The two images represent interpolations in the observation space for two observational distributions predicted from different latent codes

$\mathbf{y}_a$ and $\mathbf{y}_b$. This is shown in equation 3, where $\boldsymbol{\omega}_p \in \mathbb{R}^R$ and $\boldsymbol{\omega}_d \in \mathbb{R}^{(S \times C)}$ are both auxiliary noise variables, slerp is the spherical interpolation function (see appendix D) and $t \in [0, 1]$ is the interpolation factor.

$$\mathbf{y}_t = \boldsymbol{\mu} + \mathbf{P}\boldsymbol{\omega}_{p_t} + \sqrt{\epsilon}\,\boldsymbol{\omega}_d \tag{3}$$
$$\text{where } \boldsymbol{\omega}_{p_t} = \text{slerp}(\boldsymbol{\omega}_{p_a}, \boldsymbol{\omega}_{p_b}, t)$$

It is important to note that we only interpolate over $\boldsymbol{\omega}_p$ because it restricts the dimensionality of the hyper-sphere to size $R$; the $\sqrt{\epsilon}\,\boldsymbol{\omega}_d$ term only adds a small amount of uncorrelated noise, so setting it as a constant has negligible effect. Figure 3 uses this technique to visualise the variation contained in the observation space, demonstrating that the distribution captures semantically relevant features, such as hair colour, skin tone and background colour for the CELEBA dataset. Interpolation between differences in such features would typically entail interpolating latent variables and a forward pass through the decoder for each interval, which is not required here.

## 4.3 INTERACTIVE SAMPLING FROM THE OBSERVATION SPACE

We have demonstrated that a low-rank multivariate normal observational distribution can model a range of features. However, it would be useful if we could synthesise images with semantically meaningful human input. A step in this direction entails fixing the auxiliary noise variables associated with each sample and scaling the principal components of our covariance factor, $\mathbf{P}$. This is demonstrated in equation 4: using the singular value decomposition (SVD) of $\mathbf{P}$ and introducing a diagonal matrix of scaling coefficients, $\mathbf{A} \in \mathbb{R}^{(R \times R)}$.

$$\mathbf{P} = \mathbf{U}(\mathbf{SA})\mathbf{V}^T \tag{4}$$

Adjusting the scaling coefficients in $\mathbf{A}$ allows us to tune spatially correlated features in the sample image. The use of SVD often makes the effect of each coefficient separable and semantically relevant to the image domain. Images generated through this method for CELEBA are shown in Figure 4, where each row demonstrates the effect of scaling a different principal component. This figure shows the effect on the first ten principal components. The effect on all components and additional results on the UKBB data are given in the Appendix B. Since this manipulation is using only the observational distribution, manipulation of these samples can be achieved without performing additional forward passes on the model.

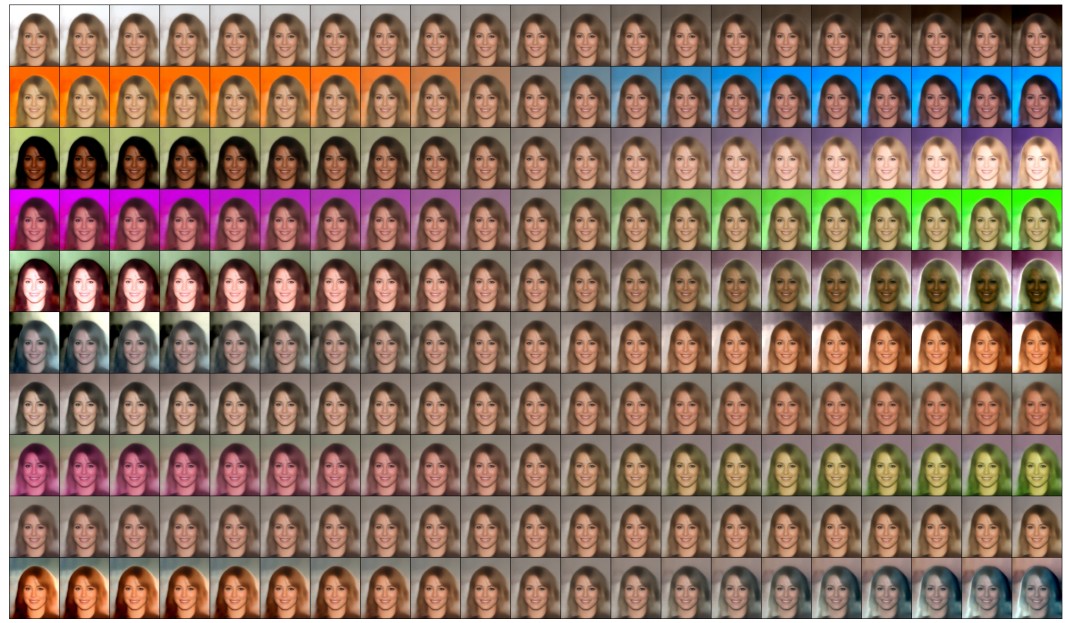

Figure 4: The effect of scaling each of the ten most principal components (each row), from top to bottom for a fixed auxiliary noise variable. The scale factor for each component ranges from $-5$ to $+5$ with intervals of $0.5$.

## 4.4 INTERACTIVE EDITING OF PREDICTIONS

One of the benefits of modelling spatial correlations in the observational distribution is that this information can be leveraged to interactively edit predictions. This involves manually editing part of the prediction and calculating the conditional distribution of the remaining pixels; for an arbitrary multivariate normal distribution, this is expressed in equations 5 and 6, where the edited pixels are modelled by $\mathbf{y}_2$ and the remaining pixels are modelled by $\mathbf{y}_1$. We use the mean of the conditional distribution ($\tilde{\boldsymbol{\mu}}$) as the corrected image. The updated covariance matrix ($\tilde{\boldsymbol{\Sigma}}$) is not needed, so we avoid evaluating it to reduce the computational cost of this process.

$$\mathbf{y} = \begin{bmatrix} \mathbf{y}_1 \\ \mathbf{y}_2 \end{bmatrix}, \quad \boldsymbol{\mu} = \begin{bmatrix} \boldsymbol{\mu}_1 \\ \boldsymbol{\mu}_2 \end{bmatrix}, \quad \boldsymbol{\Sigma} = \begin{bmatrix} \boldsymbol{\Sigma}_{11} & \boldsymbol{\Sigma}_{12} \\ \boldsymbol{\Sigma}_{21} & \boldsymbol{\Sigma}_{22} \end{bmatrix} \tag{5}$$

$$p(\mathbf{y}_1|\mathbf{y}_2 = \mathbf{b}) \sim \mathcal{N}(\tilde{\boldsymbol{\mu}}, \tilde{\boldsymbol{\Sigma}}) \tag{6}$$
$$\text{where } \tilde{\boldsymbol{\mu}} = \boldsymbol{\mu}_1 + \boldsymbol{\Sigma}_{12}\boldsymbol{\Sigma}_{22}^{-1}(\mathbf{b} - \boldsymbol{\mu}_2)$$
$$\text{and } \tilde{\boldsymbol{\Sigma}} = \boldsymbol{\Sigma}_{11} - \boldsymbol{\Sigma}_{12}\boldsymbol{\Sigma}_{22}^{-1}\boldsymbol{\Sigma}_{21}$$

Interactive editing is demonstrated in Figure 5 (with additional examples in Appendix C), where a prediction from our model, trained on the CELEBA dataset, is sequentially edited to alter the hair colour and skin tone. This demonstrates the power of the method: we can manually edit a small number of pixels and automatically update the remainder of the image coherently with the manual edit.

## 4.5 OBSERVATIONAL DISTRIBUTION WITHOUT DEEP LEARNING

Typical VAEs rely on their deep learning components to model the features for their output. In contrast, since we use a low-rank parameterisation of a full covariance matrix, our observational distribution can model spatially-correlated features on its own. As a result, the ability to model these features is not solely left to the deep learning components of the VAE; it is shared with the linear transformations that compose the low-rank multivariate normal distribution.

To understand the expressiveness of our observation space model, we carried out an experiment in which the VAE architecture is replaced with the parameters for the low-rank multivariate normal

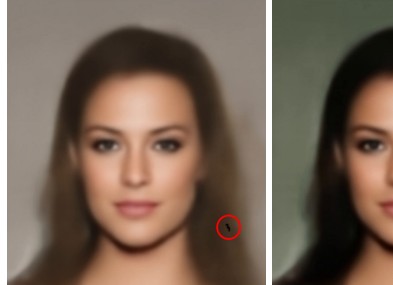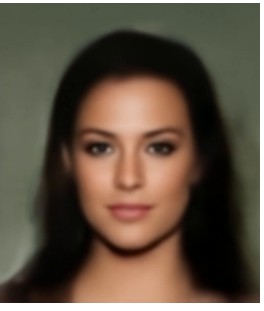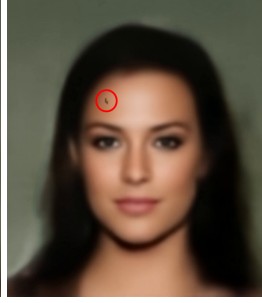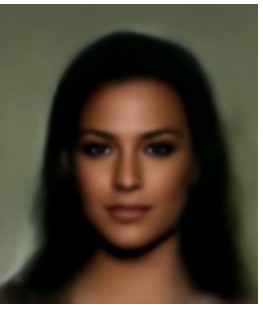

Figure 5: Sequentially editing hair colour and skin tone interactively. From left to right: a predicted image with a small coloured edit made to the hair, the image after the conditional distribution has been calculated, the image with a further edit to the skin tone, the image after the conditional distribution has been recalculated. N.B: The red circles highlighting the manual edits are for illustration purposes only and serve no computational purpose.

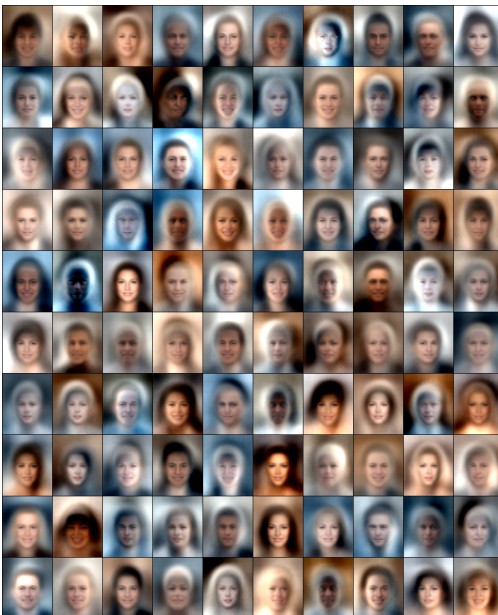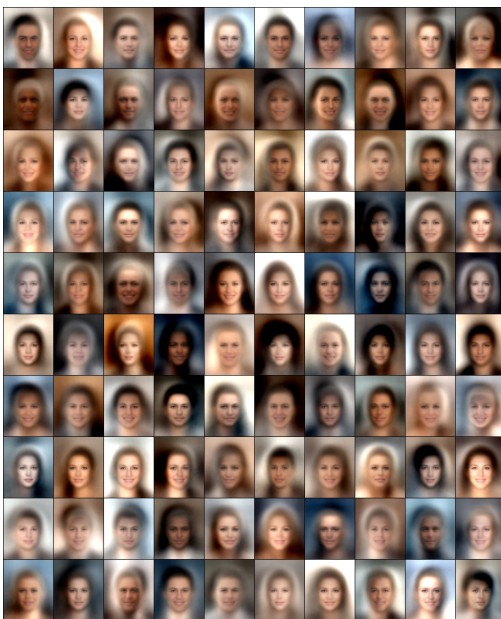

Figure 6: A qualitative comparison between 100 samples from the learnt observation space at rank = 25 with no deep learning components (left) and 100 samples from a linear PCA model with 25 features (right). In both cases, the data used for fitting is the same random subset of 10000 images from the CELEBA dataset.

distribution with no deep learning layers or latent space representation involved. We then train this simple model as described in section 3.2. This allows us to examine the capability of the model to capture the observational distribution over the dataset, without deep learning.

This modelling is reminiscent of principal component analysis (PCA), particularly given that we use a low-rank parameterisation, akin to the dimensionality reduction of PCA. Figure 6 compares samples from the distribution alongside samples from a linear PCA model of equivalent feature reduction after fitting to the CELEBA dataset (Liu et al., 2015).

There is little perceivable difference between the samples of the two methods, so we conclude that our low-rank multivariate normal distribution is comparably expressive to PCA for feature reduction. Furthermore, this experiment confirms the ability to learn the parameters of our distribution through backpropagation.

## 5 DISCUSSION

This paper highlights an often-overlooked aspect of the VAE architecture - the observational distribution. We have confirmed that pixel-wise independent observational distributions produce samples with uncorrelated pixel noise. We have introduced a low-rank multivariate normal distribution as a choice for the observational distribution of a VAE, able to model covariance between pixels and produce multiple spatially coherent samples with a single forward pass of the decoder. Our method introduces stability issues that are otherwise not present, but that we are able to resolve with an entropy constraint. Our results indicate that our choice of observational distribution is beneficial when compared to a pixel-wise independent distribution, as well as allowing sampling to be the primary method of image synthesis without reduction in quality, as theory would entail. Our method is compatible with many VAE architectures and may be applied to state-of-the-art models. We find that a low-rank multivariate observational distribution can be interpolated within, and allows for semantic, interactive manipulation of samples with a single decoder forward pass from a single latent variable.

Introducing an expressive observational distribution that is able to model features on its own, as we have, promotes a discussion comparing the features modelled in the observation space to those modelled in the latent space. In section 4.5, we observe our observational distribution's ability to model features on its own and in section 4.1 we observe a decrease in variation of the predicted means for our model compared to a standard VAE. Assuming a dataset containing finite uncertainty, we deduce that the modelling of this uncertainty is split between the latent space and the observation space. Understanding where this split lies and what influences this is an open question left for future work.

## ETHICS STATEMENT

Generative modelling is subject to dataset-inherited bias and our contributions are also susceptible. Whilst methods such as our own allow us to explore the biases that exist within a dataset, which in some cases is a desirable tool, typical usage may expose an undesired bias, particularly after training on the CELEBA dataset, which has a larger societal impact. We acknowledge that these biases, such as lack of diversity, are present but state these as artefacts of the chosen dataset and not of our own design, opinions or beliefs. There is a clear need for more diverse and representative datasets that would allow a more complete picture of the abilities and limitations of generative models to be obtained.

Our method allows for the synthesis of multiple plausible samples as well as manipulation of samples with a single forward pass of the model, where other methods require multiple forward passes. This reduction in computational burden could provide access to machine learning models for those with low-powered devices as well as reducing energy consumption and computational costs.

## REPRODUCIBILITY

All our code will be made publicly available in a dedicated GitHub repository. The CELEBA data is publicly available and we will ensure that these results are fully reproducible with the provided code. Access to the brain imaging data can be requested via a data access application to the UK Biobank Study (https://www.ukbiobank.ac.uk/).

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

# A  OUT-OF-THE-BOX SAMPLES WITHOUT STABILISING

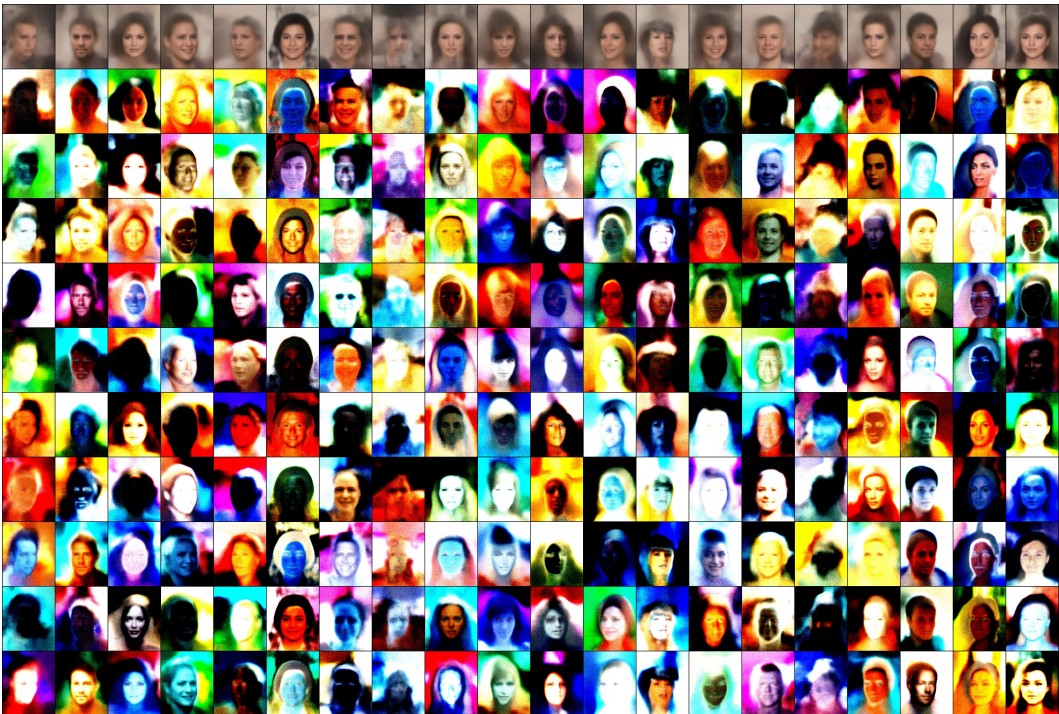

Figure 7: The results after training with a pre-training phase and weight initialisation, but without the fixed component, $\mathbf{D} = \epsilon \mathbf{I}$, or the entropy constraint. The first row represents the predicted means and all other rows represent samples from the predicted distribution outputted from the probabilistic decoder. Each column is a new sample from the latent prior decoded to predict distributions over the observation space. Model trained for 100 epochs on a random subset of 10000 images from the CELEBA dataset. Latent dimensionality: $l = 128$, rank: $R = 25$, target KL loss: $\xi_{KL} = 45$.

# B SCALING OF ALL INDIVIDUAL PCA COMPONENTS FOR CELEBA AND UKBB

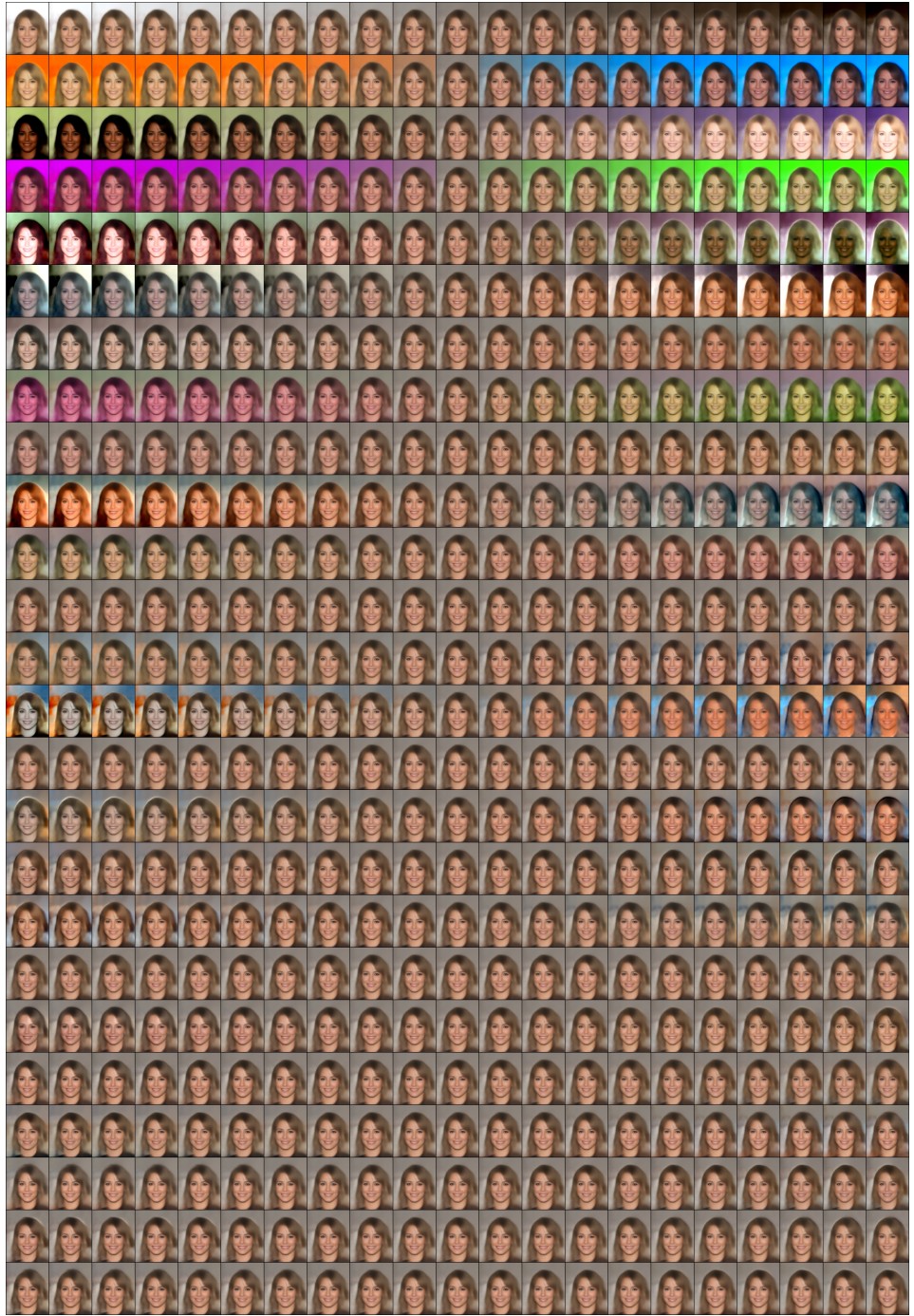

Figure 8: The effect of scaling each of the principal components (each row), from top to bottom for a fixed auxiliary noise variable and a rank 25 parameterisation. The scale factor for each component ranges from −5 to +5 with intervals of 0.5. Observational distribution predicted by our VAE after training on the CELEBA dataset.

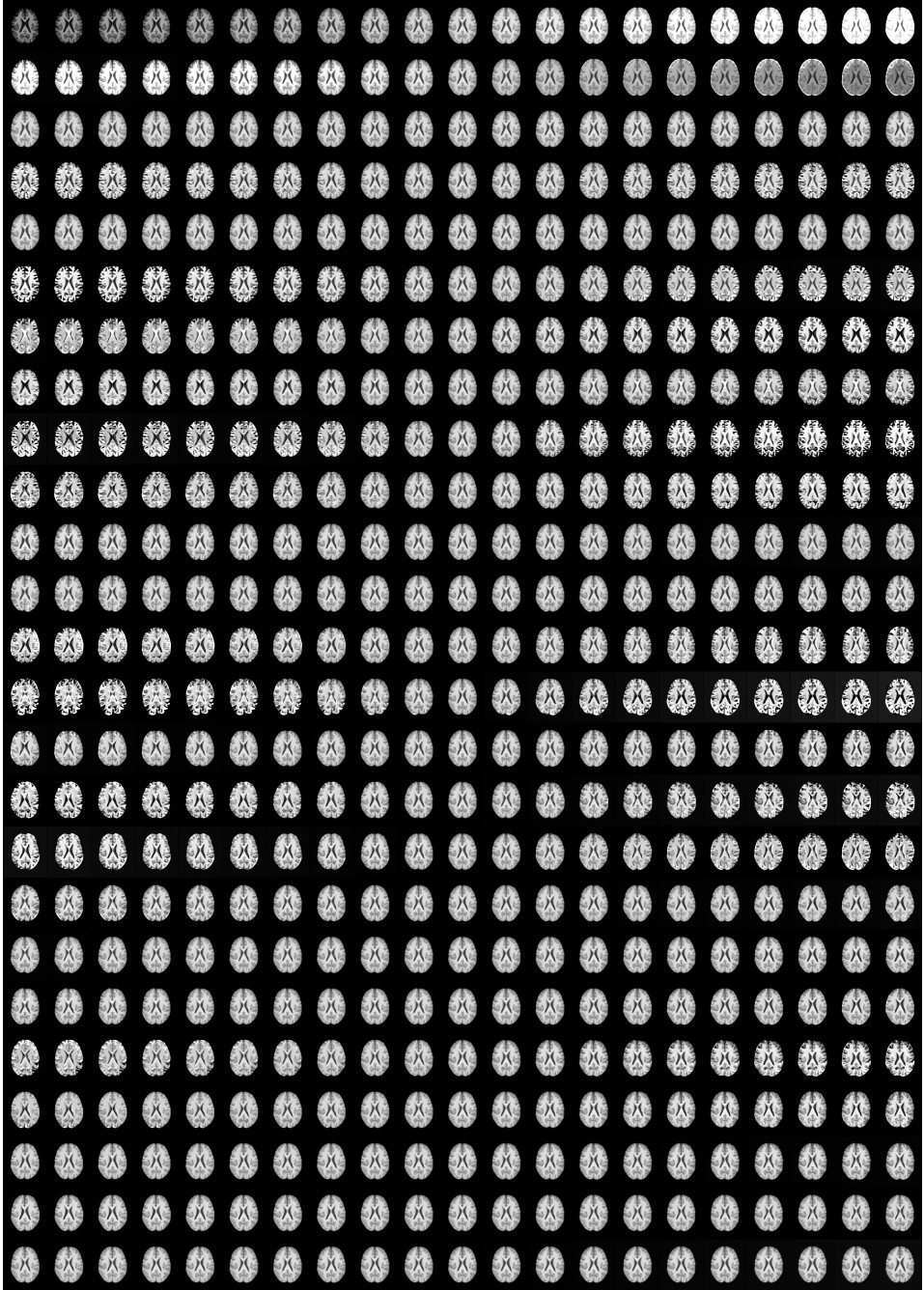

Figure 9: The effect of scaling each of the principal components (each row), from top to bottom for a fixed auxiliary noise variable and a rank 25 parameterisation. The scale factor for each component ranges from $-5$ to $+5$ with intervals of $0.5$. Observational distribution predicted by our VAE after training on the UKBB dataset.

## C    ADDITIONAL EXAMPLES FOR INTERACTIVE EDITING

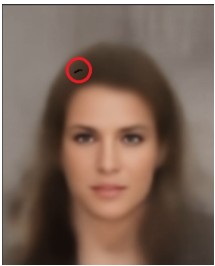 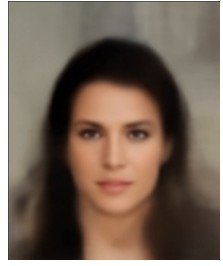 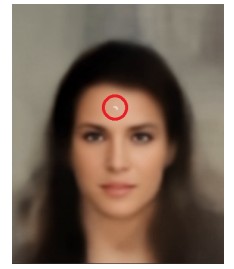 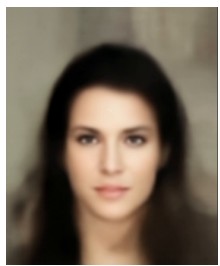

Figure 10: Sequential interactive editing. From left to right: predicted image with a small coloured edit made to the hair, the image after the conditional distribution has been calculated, the image with a further edit to the skin tone, the image after the conditional distribution has been recalculated. N.B: The red circles highlighting the manual edits are for illustration purposes only and serve no computational purpose.

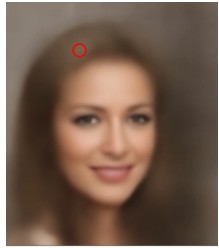 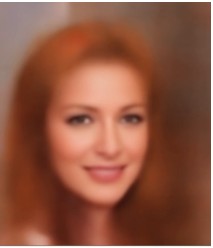

Figure 11: Interactive editing. Left: predicted image with a single-pixel manual coloured edit over the hair. Right: the mean of the calculated conditional distribution. N.B: The red circle highlighting the manual edit is for illustration purposes only and serves no computational purpose.

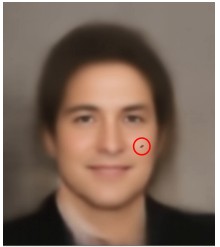 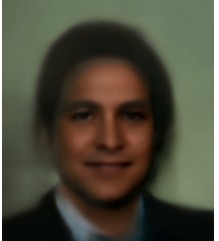

Figure 12: Interactive editing. Left: predicted image with a manual coloured edit over the skin. Right: the mean of the calculated conditional distribution. N.B: The red circle highlighting the manual edit is for illustration purposes only and serves no computational purpose.

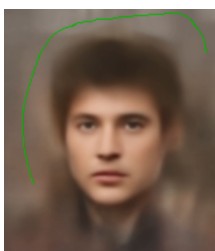 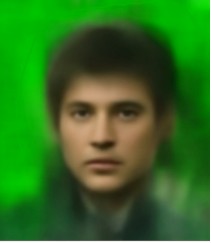

Figure 13: Interactive editing. Left: predicted image with a manual coloured edit over the background. Right: the mean of the calculated conditional distribution.

## D  SPHERICAL INTERPOLATION

For completeness, we include the spherical interpolation formula used in section 4.2.

$$\text{slerp}(\mathbf{a}, \mathbf{b}, t) = \frac{\sin((1-t)\omega)}{\sin \omega}\mathbf{a} + \frac{\sin(t\omega)}{\sin \omega}\mathbf{b} \tag{7}$$
$$\text{where} \quad \omega = \cos^{-1}(|\mathbf{a}| \cdot |\mathbf{b}|)$$

