# OpenReview forum: "Structured Uncertainty in the Observation Space of Variational Autoencoders"
_ICLR.cc/2022/Conference — ICLR 2022 Submitted_

### Official Review · Reviewer_uSoj · 2021-11-02

**Correctness:** 4
**Technical Novelty And Significance:** 3
**Empirical Novelty And Significance:** 3
**Recommendation:** 6
**Confidence:** 4

**Main Review:**

This paper is well-written, easy to follow, and well motivated. I agree with the authors that having more flexible decoder distributions, and not just more flexible neural networks (mapping to parameters of simple distributions) is an understudied area in VAEs that deserves work. The paper is very simple and relies mostly on empirical results, but I do think that the authors properly show improvements and added benefits over using diagonal covariance matrices.

One concern I have though is that it seems like adding this structure to the covariance matrix come with its share of training instabilities. The authors need to add regularizers (eq. 2), and some of the parameters used look incredibly specific (e.g. $\xi_H=-504750$ and $\xi_H=-199250$ for different datasets). How were the values of $\xi_H$ chosen? How robust are the results to changes in this hyperparameter? I think a careful answer to these questions should be presented, at least in the appendix to strengthen the paper.

Also, while I think the experiments do suggest improvements over diagonal covariance matrices, I think there are relevant additional experiments that are left out. Firstly, as the authors correctly point out, they are not the first to use a structured covariance matrix in the decoder of a Gaussian VAE [1], although this previous work uses sparse covariance matrices rather than low-rank plus diagonal ones. The absence of comparisons against this other structure significantly weakens the paper in my opinion: the baseline should not be diagonal covariance, it should be sparse covariances based on pixel neighbouring structure (as in [1]); and the reader should be convinced that low-rank plus diagonal is a more sensible choice than sparse. In other words, the authors claim "([1]) limits its ability to capture long-range spatial dependencies", and they should empirically verify that this actually hurts performance. Some more minor experimental things that I believe are also left out are: The authors should report "Our VAE (means)" in Table 1 as well. I imagine these results would be worse that the standard VAE's given for example the grey backgrounds in CELEBA, but I think this is actually interesting! It means the decoder is actually being used as a distribution in the generative model. I also believe that log-likelihoods (or ELBOs) should be reported in Table 1.

I will consider increasing my score if my main issues (hyperparameter instability, and comparisons against [1]) are adequately addressed during the rebuttal period.

Minor things:

-Citation links seem to be formatted in a different way than other papers I reviewed (they appear in blue in other papers).

-The notation S = H x W seems a bit weird: why collapse only height and width, but not channels C into a single variable? That is, S could be defined as S = H x W x C and notation would be simplified.

-Above eq.3, $t \in [0..1]$ should be $t \in [0,1]$ I believe.

-The slerp function in eq.3 should be explicitly written down, at least in the appendix.

[1] Structured uncertainty prediction networks, Dorta et al. 2018

========================================================================================================

UPDATE 1 AFTER REBITTAL

========================================================================================================

I have read the author's rebuttal, and while I understand that adding the comparisons against Dorta et al. is a significant amount of work, I am not willing to increase my score until definitive comparisons have been done, even if agree with the authors that the method of Dorta should not add large-scale noise, as ELBO/FID comparisons could remain interesting.

========================================================================================================

UPDATE 2 AFTER REBITTAL

========================================================================================================

I have read the updated author's rebuttal with comparisons against Dorta et al.'s method. I decided to increase my score, and still want to encourage the authors to include ELBO comparisons in an updated version of the manuscript.

**Summary Of The Paper:**

This paper proposes to use a low-rank plus diagonal covariance matrix, rather than the usual diagonal ones, in the decoder of Gaussian VAEs. The authors empirically show the advantages of using more expressive covariance matrices in the decoder.

**Summary Of The Review:**

While I think this paper proposes a sensible idea and shows improvements over using diagonal covariances in Gaussian VAE decoders, I have concerns about the stability of the proposed method; and I believe that the authors should compare against a stronger baseline than the one they use for the experiments to be fully convincing.

---

> ### Author Response · Authors · 2021-11-19
> **Clarifications**
>
> We thank you for your review, which is thoughtful and well-considered. We agree that flexible decoder distributions are important and understudied, and we hope that our paper will help convince the community of this.
>
> Regarding the stability of our method, optimisation of variance networks is known to be unstable [1]. To address this, we use the modified differential method of multipliers to optimise our Lagrangian [2]. This method allows for convergence and stability in constrained optimisation problems such as the VAE objective.
>
> Regarding the choice of slack variables, please refer to our response in the joint reply.
>
> As suggested, we included (pending) the following additional experiments:
> Comparison with Dorta et al. [3] in Table 1;
> Comparison with ‘our VAE (means)’;
> ELBO values included in Table 1.
>
> We have also changed the formatting of the citation links to bring them in line with your suggestion, added the slerp equation to the appendix, and changed the formatting above equation 3. We appreciate your point about the S = HxW notation, although we have left it unchanged to keep it consistent with the notation used in Monteiro et al. [4].
>
> We thank you for the care you have taken in your review and hope that our response has been satisfactory. If there are any significant points left unaddressed, we would be happy to continue this discussion.
>
>
> References:
>
> [1] Andrew Stirn and David A. Knowles.  Variational variance: Simple and reliable predictive variance parameterization. CoRR , abs/2006.04910, 2020.  URL https://arxiv.org/abs/2006.04910.
>
> [2] John Platt and Alan Barr. Constrained   differential   optimization. In   D.   Anderson    (ed.), Neural  Information  Processing  Systems. American Institute of Physics, 1988. URL https://proceedings.neurips.cc/paper/1987/file/a87ff679a2f3e71d9181a67b7542122c-Paper.pdf
>
> [3] Dorta G, Vicente S, Agapito L, Campbell ND, Simpson I. Structured uncertainty prediction networks. In Proceedings of the IEEE Conference on Computer Vision and Pattern Recognition 2018 (pp. 5477-5485).
>
> [4] Monteiro M, Folgoc LL, de Castro DC, Pawlowski N, Marques B, Kamnitsas K, van der Wilk M, Glocker B. Stochastic segmentation networks: Modelling spatially correlated aleatoric uncertainty. Neurips 2020.

---

> > ### Comment · Reviewer_uSoj · 2021-11-22
> > **Clarifications reply**
> >
> > I thank the authors for their response to my review. I noticed that while Table 1 now has "Our VAE (means)" values, ELBO values and comparisons against the method of Dorat et al. are still missing. I understand these experiments can take time, but I will keep my score in the absence of these additional comparisons.

---

> > > ### Author Response · Authors · 2021-11-23
> > > **Please see important update at the top**
> > >
> > > Many thanks for the feedback on the rebuttal. We have just posted an update on the experiments with Dorta et al.'s method at the top and we would be grateful if you would take this into account when reassessing our paper.

---

> > > > ### Author Response · Authors · 2021-11-30
> > > > **Follow-up**
> > > >
> > > > We were wondering if the reviewer had a chance to have a look at our response and also the additional clarifications and updates posted at the top. Did this address your previous concerns? Many thanks for taking the time.

---

> > > > > ### Author Response · Authors · 2021-12-09
> > > > > **New results**
> > > > >
> > > > > Please note our new results regarding the comparison with Dorta et al.'s method posted at the top.

---

### Official Review · Reviewer_DqfM · 2021-11-07

**Correctness:** 2
**Technical Novelty And Significance:** 2
**Empirical Novelty And Significance:** 3
**Recommendation:** 3
**Confidence:** 4

**Main Review:**

More quantitative results are needed to demonstrate possible advantages of the proposed method. Overall the only contrubution of this paper is to transfer the method of Monteiro et al. into the variational autoencder framework. Therefore this paper should clearly demonstrate the benefits of the proposed method over the established practice of using the predicted mean output vector of the decoder.
Fig. 4 a) and b) compare the proposed method with the established practice of using the predicted mean output vector of the decoder. However in Fig. b) the samples from the proposed method shows some unfavourable color shift. Thus the mean vector creates the more realistic image and would still be the preferable option.

More quantitative results are needed:

- "Structured uncertainty": Is the uncertainty calibrated? Please provide experiments on synthetic data, for example sampled from a multivariate distribution with known mean and variance to demonstrate that the framework actually models the uncertainty of the data distribution.
- Use of constraints in Eq. 1: Is convergence of the model affected? Are the Lagrangian multupliers maximized during optimization? Does this affect the numerical stabiilty during training?
- How were the dataset-specific slack variables in Eq, 1 chosen?

References

[Monteiro at al. 2020] Monteiro, Miguel, et al. "Stochastic segmentation networks: Modelling spatially correlated aleatoric uncertainty, Stochastic Segmentation Networks: Modelling Spatially Correlated Aleatoric Uncertainty, Neurips 2020


**Summary Of The Paper:**

In the standard Variational Autoencoder framework the statistics of the decoder output are assumed to be pixel-independent Gaussian which can lead to problems when sampling from the model when covariances are missing. To overcome these limitations, the authors propose to use the network architecture proposed by Monteiro at al. 2020 in the decoder of a variational autoencoder. In the approacah of
Monteiro at al. 2020, the output distribution is modeled as low-rank multivariate normal. Qualitative results on the CelebA dataset demonstrate that samples from the proposed modified variational autoencoder capture covariances between different parts of the image.

**Summary Of The Review:**

The contribution of this paper is limited, it merely takes an existing method to capture covariances of model outputs and incoroporates it into the decoder of a variational autoencoder. More quantitative results are needed to demonstrate possible advantages of the proposed method and calibration of the estimated covariances. Further details on parameter optimization and convergence are needed.

---

> ### Author Response · Authors · 2021-11-19
> **Clarifications**
>
> Thank you for taking the time to review our paper. It seems your main concern was our use of qualitative instead of quantitative results. As mentioned in the joint reply, we have reimplemented Dorta et al. [1] from the related work and compared our results numerically, using both FID scores and ELBO scores. We believe this significantly strengthens our paper and hope this addresses your concern.
>
> We would like to ask for clarification regarding your comment questioning if the uncertainty is calibrated. How would we measure uncertainty calibration without making incorrect assumptions of independence between pixels? Since the true joint distribution over images is unobtainable, we cannot measure the uncertainty calibration on image datasets. Experiments on synthetic data would amount to fitting a normal distribution to samples from a normal distribution. Since we believe this to be a trivial experiment/result we did not include it.
>
> Regarding convergence and stability of the constrained optimisation objective. In contrast with balancing the two terms in the VAE objective with a fixed scalar, using the modified differential method of multipliers to optimise our Lagrangian guarantees convergence and does not impact stability [2].
> As per the modified differential method of multipliers [2]: the Lagrangian multipliers are maximised while the constraint is not satisfied and minimised once it is satisfied.
>
> Regarding the choice of slack variables, please refer to our response in the joint reply.
>
> We also wish to respond to your comment: “...the samples from the proposed method shows some unfavourable colour shift. Thus the mean vector creates the more realistic image and would still be the preferable option.” A low-rank observational distribution allows modelling a coherent distribution over images with a single forward pass. This distribution naturally captures properties of the data, such as colour balance. Thus, random samples will display diverse colour balances if observed in the training data. This property is not a bug. It is a feature of the expressivity of the low-rank multivariate normal distribution and the data used for training. Since CELEBA is a photography dataset with a range of diverse colour palettes this "colour shift" is expected and desirable. We also show in Figure 4 and Figure 5 that we can interactively control these characteristics.
>
> We hope that our responses have helped to demonstrate the strengths of our work. Finally, we ask if you could clarify your correctness score of 2 - “Several of the paper’s claims are incorrect or not well-supported.” - we are not sure which claims you believe are incorrect? We thank you for your time and your consideration and look forward to continuing this discussion.
>
> References:
>
> [1] Dorta G, Vicente S, Agapito L, Campbell ND, Simpson I. Structured uncertainty prediction networks. In Proceedings of the IEEE Conference on Computer Vision and Pattern Recognition 2018 (pp. 5477-5485).
>
> [2] John Platt and Alan Barr. Constrained   differential   optimization. In   D.   Anderson    (ed.), Neural  Information  Processing  Systems. American Institute of Physics, 1988. URL https://proceedings.neurips.cc/paper/1987/file/
> a87ff679a2f3e71d9181a67b7542122c-Paper.pdf

---

> > ### Author Response · Authors · 2021-11-30
> > **Follow-up**
> >
> > We were wondering if the reviewer had a chance to have a look at our response and also the additional clarifications and updates posted at the top. Did this address your previous concerns? Many thanks for taking the time.

---

> > > ### Author Response · Authors · 2021-12-09
> > > **New results**
> > >
> > > Please note our new results regarding the comparison with Dorta et al.'s method posted at the top.

---

### Official Review · Reviewer_cypB · 2021-11-08

**Correctness:** 4
**Technical Novelty And Significance:** 2
**Empirical Novelty And Significance:** 2
**Recommendation:** 5
**Confidence:** 4

**Main Review:**

The problem the authors tackle in this paper and the proposed solution are interesting. The submission is technically sound and the paper is well written.

However, I have some major concerns about the originality of the work and feel that further comparison with other methods would be needed to show improvements over the state-of-the-art methods. Specifically:
1) this work is closely related to Monteiro et al. (2020), who used a similar model in a segmentation setting. Is the model proposed here the same considered in Monteiro et al? If not, the authors should highlight methodological differences. If yes, the authors should highlight the conceptual contributions of their work.
2) Dorta et al. (2018) considered a closely-related model, but making a different modeling choice for the covariance matrix. The authors should compare their results with those obtained with the model from Dorta et al (e.g., comparison of FID scores of generated samples).

Other questions / concerns:
- I find it hard to judge whether the means from the structured-observation-space VAE look visually more realistic than the mean from the canonical VAE. Can the author comment on this? Also, does not "fixing the covariance diagonal to a small positive scalar" artifactual reduce independent noise in the samples? I wonder if a visual comparison with canonical VAE is fair in this setting.
- Are there other potential use cases for the observational model beyond pixel editing?
- I think another way to improve upon the canonical VAE likelihood is to consider an iid gaussian likelihood on perceptual features [1]. The authors should consider this method in the related method section and potentially include this technique in the comparison.

[1] Hou X, Shen L, Sun K, Qiu G. Deep feature consistent variational autoencoder. In2017 IEEE Winter Conference on Applications of Computer Vision (WACV) 2017 Mar 24 (pp. 1133-1141). IEEE.

**Summary Of The Paper:**

The authors aim at improving the canonical VAE model by replacing the standard iid Gaussian likelihood with a multivariate Gaussian with (low-rank + diagonal) covariance.

In applications to CelebA and a brain MRI dataset from UK Biobank, the authors compare the proposed structured-observation-space VAE with a canonical VAE, showing that the samples from their model have lower Fréchet Inception Distance (FID) scores than both means and samples from the canonical VAE. The authors then evaluate the expressiveness of the representations learned in the observation space by visually evaluating interpolations in the observation space, and images obtained by rescaling the principal components of the observational covariance matrix. Finally, the authors show how their model can be used for interactive editing—i.e., editing a small number of pixels and using the conditional distribution to infer coherent edits in the remainder of the image.

**Summary Of The Review:**

The paper is sound, well written and tackles an interesting problem. However, more work is needed to demonstrate improvement upon state of the art methods (specially, Dorta et al, 2020, and potentially Hou et al, 2017).

---

> ### Author Response · Authors · 2021-11-19
> **Clarifications**
>
> Thank you for your detailed review which has been helpful to improve our paper. You raise two main concerns, both of which we address in our joint response, but we will elaborate upon here:
>
> 1. Although our work parameterized the covariance matrix in the same style as Monteiro et al. [1], we would like to stress that this is the first work using such a parameterization in generative modelling, which is novel and non-trivial. Ultimately, we hope that our work will draw attention to the observational distribution used in VAEs and spur a conversation in the generative modelling community about modelling features with the decoder distribution. Furthermore, to address specific challenges in generative modelling, we propose an additional constraint and use of a Lagrangian optimization, which are not present in [1].
>
> 2. We have implemented the method from Dorta et al. [2] and included a comparison and discussion. We believe this substantially strengthens our paper and hopefully addresses your main concerns.
>
> You also raise three further questions:
> - Comparing the means of our structured observation space VAE with the means of the canonical VAE is indeed interesting. Qualitatively, we observe less variation in our means (e.g. the gray backgrounds for CELEBA). This effect can be explained by the ability to capture more uncertainty in the observation space. Thus, compared to the canonical VAE, we observe less uncertainty in the latent space since it has been "moved" to the observation space. As for visual quality, this is unfortunately subjective. In our amended version of the paper we also report the FID for our means which may help to answer this. However, we would like to emphasize that, whilst in the canonical VAE only the means are coherent synthesised images, in our approach both the mean and samples are plausible.
> - There are multiple use cases for our observational distribution. By using a single forward pass of the decoder to obtain a distribution over images, we foresee its usefulness applications including but not limited to:
>   - On-the-fly data augmentation (e.g., in self-supervised learning);
>   - Efficient conditional sampling;
>   - Efficient uncertainty estimation;
>   - Multi-output prediction (potentially with a human-in-the-loop).
>
> - We agree that Hou et al. [3] is related, and we have updated our manuscript to reflect this. Note, they use a pretrained VGG network to enforce pixel similarity and lack any probabilistic interpretation of the VAE. There is no likelihood in pixel space, setting their method clearly apart from our work.
>
> We thank you for your time and are happy to continue this conversation if there are any remaining concerns or points of clarification.
>
> References:
>
> [1] Monteiro M, Folgoc LL, de Castro DC, Pawlowski N, Marques B, Kamnitsas K, van der Wilk M, Glocker B. Stochastic segmentation networks: Modelling spatially correlated aleatoric uncertainty. Neurips 2020.
>
> [2] Dorta G, Vicente S, Agapito L, Campbell ND, Simpson I. Structured uncertainty prediction networks. In Proceedings of the IEEE Conference on Computer Vision and Pattern Recognition 2018 (pp. 5477-5485).
>
> [3] Hou X, Shen L, Sun K, Qiu G. Deep feature consistent variational autoencoder. In2017 IEEE Winter Conference on Applications of Computer Vision (WACV) 2017 Mar 24 (pp. 1133-1141). IEEE.

---

> > ### Author Response · Authors · 2021-11-30
> > **Follow-up**
> >
> > We were wondering if the reviewer had a chance to have a look at our response and also the additional clarifications and updates posted at the top. Did this address your previous concerns? Many thanks for taking the time.

---

> > ### Comment · Reviewer_cypB · 2021-11-30
> > **Possible to share results from comparison with Dorta et al?**
> >
> > I really want to be supportive of this paper but it is hard for me to do so without seeing a comparison with Dorta et al. All reviewers raised the same point. This seems to be a major weakness of this work.
> >
> > Apologies if I missed anything here, but can the author present some of the current results of their comparison in the response? (The PDF does not have the additional results; I think they cannot be added there, at this stage)
> >
> > Specific comments on their response:
> >
> > > We reiterate that our low-rank parameterization of the covariance matrix can capture global spatial correlations, in contrast to the local band-diagonal approach of Dorta et al
> >
> > I do not think this is correct. Dorta et al consider a tridiagonal form for the precision matrix, not the covariance. A precision matrix specifies conditional independencies; its inverse (the covariance) can still capture global correlations. Their modeling choice makes a lot of sense, actually.
> >
> > > Although our work parameterized the covariance matrix in the same style as Monteiro et al. [1], we would like to stress that this is the first work using such a parameterization in generative modeling
> >
> > Technically, there are not many implementation differences between a supervised model for segmentation and a VAE. However, I do agree that drawing attention to this issue in the generative modeling community is important, which may be enough to address my previous comment.
> >
> > > We have implemented the method from Dorta et al. [2] and included a comparison and discussion
> >
> > Could you please point me to these results or describe them here?
> >
> > > whilst in the canonical VAE only the means are coherent synthesised images, in our approach both the mean and samples are plausible
> >
> > Of course, but empirically, no one considers samples from a VAE. I do not want to be overly critical with this comment, but I think it is worth noticing this as it is the underlying reason why showing better samples may not be enough (empirically) for a comparison. I do see that you also show better FID for the means.
> >
> > I do wonder if an image imputation task may help address this even more convincingly.
> >
> > > We agree that Hou et al. [3] is related, and we have updated our manuscript to reflect this. Note, they use a pretrained VGG network to enforce pixel similarity and lack any probabilistic interpretation of the VAE. There is no likelihood in pixel space, setting their method clearly apart from our work.
> >
> > I agree. Worth adding to the discussion though as still related.

---

> > > ### Author Response · Authors · 2021-12-02
> > > **Use of samples**
> > >
> > > Thanks for getting back, very much appreciated.
> > >
> > > - Regarding the comparison with Dorta et al. please see our latest update on the main thread. So far we have been struggling with some practical limitations of their approach. We believe this is important to consider when assessing our overall contribution.
> > >
> > > - We agree, the global vs local argument is not accurate. Thanks for pointing this out.
> > >
> > > - Regarding the use of samples you said ‘no one considers samples from a VAE' which is exactly the point we are addressing. As the majority of works assume a pixel-wise independent distribution in the observation space, the samples are noisy, implausible, and not useful. With our method it is possible to consider the samples, not just the mean. From a theoretical point of view, a generative model should allow drawing plausible samples if the underlying distribution is faithfully represented. So this is not just a nuisance of the model. From a practical point of view, our method allows the generation of multiple, plausible predictions with a single forward pass which is relevant for practical applications including the ones listed above (on-the-fly data augmentation, efficient conditional sampling and uncertainty estimation, multi-output prediction).

---

> > > ### Author Response · Authors · 2021-12-09
> > > **New results**
> > >
> > > Please note our new results regarding the comparison with Dorta et al.'s method posted at the top.

---

### Official Review · Reviewer_2FaS · 2021-11-08

**Correctness:** 3
**Technical Novelty And Significance:** 2
**Empirical Novelty And Significance:** 2
**Recommendation:** 6
**Confidence:** 4

**Main Review:**

Strengths
- The authors provide an adequate introduction and summary of the literature, and highlight how their proposed method could be plugged into existing models.

Weaknesses:
- The approach is not radically novel and the paper could improve further by providing all details to allow the reader to fully reproduce their results. This could be included in an extended supplementary section.
- Another part that could greatly improve the paper is some analysis of the scale of the variables.
- The paper would greatly benefit of analysis showing how the approach extended into a more complex VAE framework further improve any downstream task.
- How should the reader interpret the scales of the Lagrangian multiplier and the slack variable in Eq (2)? In section 4.1 these values are reported for (2) datasets. Adding additional analysis/explanation for these variables would benefit the paper and help persuade any reader about the robustness of  the results presented.
- Figure 2 would also benefit from further explanation to better understand how these results support the claim of the authors.

**Summary Of The Paper:**

The paper proposes to include some structure in the observation model of the standard VAE, typically implemented with a normal distribution with diagonal covariance matrix. The authors follow the work in [Monteiro 2020] which include low rank structure structure in the covariance matrix for the observation model in another generative framework.


**Summary Of The Review:**


The paper provides an extension of the work in [Monteiro et al. 2020] to the case of VAEs. The paper provides adequate references but it would greatly benefit for more downstream experiments and detailed explanation as suggested in the main review.

---

> ### Author Response · Authors · 2021-11-19
> **Clarifications**
>
> Thank you for your constructive review, which raises some interesting questions. We will address each one sequentially:
>
> - We address the concern regarding novelty in the joint reply. We wish to emphasize that the choice of observational distribution in VAEs is an understudied area and believe our paper can bring some attention to it. Our code will be made available on GitHub for full reproducibility of all results, and to facilitate future work in this direction. You asked for details in the supplementary material, but we are unsure if this would anything in addition to the code release. Were there any specific details you would like use to add in addition to the code (which will come with instructions how to use it)?
>
> - Regarding the slack variables, scale and use of Lagrangian multipliers, please see the point 3 in the joint reply.
>
> - Regarding more complex VAE frameworks, we would first like to emphasize that any VAE that models a pixel-independent distribution, no matter the architecture, is susceptible to the problem we are addressing with our method. Most current VAE frameworks suffer from this issue. Our approach is complementary and can be implemented with minimal changes and computational cost to the underlying model. It also enables other frameworks to benefit from the applications we highlight in the paper, including spatially coherent samples, multiple samples with a single forward pass, observation space interpolation and image editing.
>
> - We thank you for suggesting points for improving the clarity of our presentation. We have amended the paper to add extra discussion of Figure 2. We have now added that rows 5 and 6 are the positive and negative covariance to the central pixel.
>
> Thank you for your time and your insightful comments. We hope to continue this discussion and to clarify any further questions you may have.

---

> > ### Author Response · Authors · 2021-11-30
> > **Follow-up**
> >
> > We were wondering if the reviewer had a chance to have a look at our response and also the additional clarifications and updates posted at the top. Did this address your previous concerns? Many thanks for taking the time.

---

> > > ### Author Response · Authors · 2021-12-09
> > > **New results**
> > >
> > > Please note our new results regarding the comparison with Dorta et al.'s method posted at the top.

---

### Author Response · Authors · 2021-11-19
**Joint reply addressing common points and highlighting our contribution**

We would like to thank the reviewers for their time and their detailed comments. We will reply to each reviewer individually to clarify any specific questions and provide details how we addressed them; The reviews were very helpful for improving the paper and strengthening our contributions. We would like to ask the reviewers to consider our responses and reconsider their recommendations in the light of these.

In this general reply, we want to address three common points as we felt it is important to highlight these for the reconsideration of the reviewers and recommendation by the area chairs.

1. It was rightfully stated that a comparison with the method by Dorta et al. [1] would add significant value to our paper. We have now reimplemented their method and the (pending) results are added to our paper (Table 1) as a point of comparison with state-of-the-art. We reiterate that our low-rank parameterization of the covariance matrix can capture global spatial correlations, in contrast to the local band-diagonal approach of Dorta et al - as demonstrated the low-rank structure captures global semantically relevant information such as hair color, background color, and skin tone in the CELEBA model. We thank the reviewers for the suggestion, and we believe the additional experiments and comparison further strengthens our manuscript.

2. It was further criticized that our methodological contribution is similar to Monteiro et al. [2] who use a low-rank parameterization of the covariance matrix to model aleatoric uncertainty in discriminative models in the context of semantic segmentation. We disagree with the sentiment of limited novelty as our paper makes clear contributions in the context of generative models. While our method shares the same parameterization, there is substantial value and novelty in our work which bridges across different areas. We show that using a structured observational distribution in generative models can improve the fidelity of the generated images, whilst also providing tools to demonstrate how the images can be interactively manipulated in the observation space without needing the model. Our use of a more expressive distribution in the decoder to capture relevant semantic information pushes back against a recent trend of simply using larger networks at scale and demonstrates that focusing attention on this understudied area can lead to significant improvements in VAEs. This has not been done before, and our work may inspire future directions in generative modelling as we believe the choice of observational distribution to be an under-studied area. We posit that our contributions are relevant, insightful, and impactful for the generative modelling community and thus meet the threshold for ICLR.

3. Regarding the choice of slack variables, one of the benefits of Lagrangian optimization is that these variables have a semantic meaning. Nonetheless, obtaining specific values was done empirically through observation of training set samples. The resulting slack values can be interpreted to make sure they are within sensible ranges for their definition. This cannot be done with a standard beta VAE where the beta parameter has no scale.

We once again thank the reviewers for their careful reviews and for engaging with us to help improve our paper.

References:

[1] Dorta G, Vicente S, Agapito L, Campbell ND, Simpson I. Structured uncertainty prediction networks. In Proceedings of the IEEE Conference on Computer Vision and Pattern Recognition 2018 (pp. 5477-5485).

[2] Monteiro M, Folgoc LL, de Castro DC, Pawlowski N, Marques B, Kamnitsas K, van der Wilk M, Glocker B. Stochastic segmentation networks: Modelling spatially correlated aleatoric uncertainty. Neurips 2020.

---

### Author Response · Authors · 2021-11-23
**IMPORTANT UPDATE: Difficulties to obtain convincing results with Dorta et al.’s method**

We have been working very hard on the implementation of Dorta et al.’s method over the last week, but we have been struggling to obtain convincing results for our experimental setup with color images of higher resolution (3x218x178). At this stage, we cannot with certainty conclude if there is a more fundamental problem with their method when applied to larger images with multiple channels. Note that their original paper only showed results on single channel grayscale images of size 1x64x64.

While we have been able to obtain somewhat reasonable mean predictions with their method, suggesting that our implementation is working and the model is training well, the generated samples, however, show a high level of unstructured noise. The quality and plausibility of the samples is nowhere near to the samples obtained with our method. This may indicate that the sparse parameterization in Dorta et al. is indeed unable to capture long-range spatial dependencies that are required to obtain realistic samples as we argued in our paper. In fact, when closely inspecting again the visual results in the original paper, one can observe some unstructured noise in their samples (cf. Fig. 7 where the authors call this “plausible high-frequencies”). What may have been visually plausible on the low-res images could actually be problematic on the more complex data that we consider in our paper.

Given the little time we had available for the implementation and for conducting these experiments (each training run takes about two days), we hope the reviewers will understand that we do not yet feel comfortable to put these results in our revised paper as we would like to make sure they are valid and representative of the performance of Dorta et al.’s method. Here, we would like to assure the reviewers that a fair comparison with Dorta et al. will be included in the final version, which addresses a key concern. Here, we would like to highlight again the key differences between a sparse, band-diagonal model as proposed by Dorta et al. and our low-rank, global parameterization which offers a novel, effective and efficient modelling choice for the observational distribution of generative models.

---

### Author Response · Authors · 2021-12-02
**Further update regarding a comparison with Dorta et al.**

We have continued our efforts of reimplementing the method by Dorta et al. to provide the requested comparative evaluation. However, we have run into practical issues when implementing their method for RGB images. To initially confirm that our base implementation is working, we tested it using the image’s natural pixel ordering (row-wise) to build the covariance, however, this naturally produces correlated pixels in rows instead of blocks as intended by Dorta et. al. Using a block structure over the image data requires substantial engineering efforts to overcome memory constraints. A naive implementation replicating pixels for every block poses computational issues.

At the same time, we attempted to directly use Dorta et al.’s original implementation (GitHub: [https://github.com/Garoe/tf_mvg](https://github.com/Garoe/tf_mvg)) but unfortunately the provided code only supports grayscale images (which is [acknowledged in their provided examples](https://github.com/Garoe/tf_mvg/blob/01bc681a8b3aac5dcf0837d481b963f4968eb777/examples/autoencoder_mvg_chol_filters.py#L32)). Extending their code to RGB is non-trivial. Additionally, their implementation is not vectorized over the batch dimension (it uses a for loop) which significantly limits the image and batch size. As a consequence, a quantitative comparison on the setting we considered in our paper (RGB, higher resolution) between our method and Dorta et al. is currently not feasible. We are now aiming for a quantitative comparison on grayscale images of size 64x64.

We would like to ask the reviewers and area chairs to take these practical issues of Dorta et al.’s method into account when judging about our contributions. Our method does not suffer from these problems, and we believe that the practicality of a method is an important aspect of the overall contribution.

---

### Author Response · Authors · 2021-12-09
**Results now available for Dorta et al. showing clear improvement for our method**

We are very pleased to share the results for the requested comparison with Dorta et al.'s method demonstrating better quantitative and qualitative performance for our proposed approach.

We report the FID metric calculated over 50,000 means and 50,000 samples generated with the three compared models (Standard VAE, Dorta, Ours) trained on the CELEBA dataset.

- Standard (means) : 90.97
- Dorta    (means)    : 73.44
- Ours     (means)    : 70.43


- Standard (samples) : 257.11
- Dorta    (samples)    :  77.72
- Ours     (samples)    :  74.66

We observe improvements in FID both for the means and the samples. Additionally, we provide a visual comparison of generated samples showing that our samples are of much higher quality: [https://i.imgur.com/k8UdH0Q.png](https://i.imgur.com/k8UdH0Q.png)

Note, this comparison was done on grayscale images as the code provided by Dorta et al. does not support color images. We would also like to highlight that we had to extend their code to VAEs ourselves which took longer than expected, as their provided examples are limited to autoencoders. We will share our VAE implementation of Dorta et al.'s method as part of our code release.

We hope these results convince the reviewers of the value of our method, and that this addresses their previous concerns. We would be grateful if the reviewers would reconsider their previous recommendations in the light of these results which will be added to the paper.

---

### Decision · Program_Chairs · 2022-01-20

**Decision:**

Reject

**Comment:**

In order to evaluate the evidence lower bound (ELBO), VAEs typically use a parametric distribution-based decoder $p(x|z)$. If the data is continuous, one often considers a Gaussian VAE, where the canonical setting is to assume a diagonal covariance matrix $p(x|z) = N(x; \mu(z), \sigma^2 \mathbf{I})$. In this paper, the authors suggest replacing the diagonal covariance matrix with a structured covariance matrix (low-rank + diagonal). As this only amounts to a minor change to a canonical Gaussian VAE, strong empirical results are expected to justify its acceptance. However, the image generation results presented in the paper are not comparable to the state-of-the-art VAE results (e.g., Arash Vahdat, and Jan Kautz. "NVAE: A Deep Hierarchical Variational Autoencoder." Neural Information Processing Systems (NeurIPS), 2020).